# Metagenomics of Neotropical Single-Stranded DNA Viruses in Tomato Cultivars with and without the *Ty*-1 Gene

**DOI:** 10.3390/v12080819

**Published:** 2020-07-28

**Authors:** Luciane de Nazaré Almeida dos Reis, Maria Esther de Noronha Fonseca, Simone Graça Ribeiro, Fernanda Yuri Borges Naito, Leonardo Silva Boiteux, Rita de Cássia Pereira-Carvalho

**Affiliations:** 1Departamento de Fitopatologia, Universidade de Brasília (UnB), Área de Virologia Vegetal, Brasília-DF 70910900, Brazil; lucy_reis_@hotmail.com (L.d.N.A.d.R.); naitofyb@gmail.com (F.Y.B.N.); 2National Center for Vegetable Crops Research (CNPH), Embrapa Hortaliças, Brasília-DF 70275-970, Brazil; maria.boiteux@embrapa.br; 3Embrapa Recursos Genéticos e Biotecnologia, Brasília-DF 70770-917, Brazil; simone.ribeiro@embrapa.br

**Keywords:** begomoviruses, tomato, resistance gene, NGS, virome

## Abstract

A complex of begomoviruses (*Geminiviridae*) can cause severe tomato yield losses in the neotropics. Here, next-generation sequencing was employed for large-scale assessment of single-stranded (ss)DNA virus diversity in tomatoes either harboring or lacking the large-spectrum begomovirus tolerance *Ty*-1 gene. Individual leaf samples exhibiting begomovirus-like symptoms (*n* = 107) were field-collected, circular DNA-enriched, subdivided into pools (with and without *Ty*-1), and Illumina-sequenced. Virus-specific PCR and Sanger dideoxy sequencing validations confirmed 15 distinct ssDNA virus/subviral agents (occurring mainly in mixed infections), which highlight the potential drawbacks of employing virus-specific resistance in tomato breeding. More viruses (14 versus 6 species) were observed in tomatoes without the *Ty*-1 gene. A gemycircularvirus (*Genomoviridae*), a new alpha-satellite, and two novel *Begomovirus* species were identified exclusively in samples without the *Ty*-1 gene. A novel begomovirus was found only in the *Ty*-1 pool, being the only species associated with severe symptoms in *Ty*-1 plants in our survey. Our work is the first step towards the elucidation of the potential begomovirus adaptation to *Ty*-1 and its specific filtering effects on a subset of ssDNA viral/subviral agents.

## 1. Introduction

*Geminiviridae* is the largest family of plant-infecting viruses, and is currently organized into nine genera: *Becurtovirus*, *Begomovirus*, *Capulavirus*, *Curtovirus*, *Eragrovirus*, *Grablovirus*, *Mastrevirus*, *Topocuvirus*, and *Turncurtovirus* [1]. The classification at the genus level is based upon host range, the associated insect vector(s), genomic organization, and phylogenetic relationships [1,2,3]. In 2016, two novel viral families with non**-**enveloped, circular, single-stranded DNA (ssDNA) genomes with sizes ranging from 2.0 to 2.4 kb were created and named as *Pleolipoviridae* and *Genomoviridae*. The *Genomoviridae* family also comprises nine genera: *Gemycircularvirus*, *Gemyduguivirus*, *Gemygorvirus*, *Gemykibivirus*, *Gemykolovirus*, *Gemykrogvirus*, *Gemykroznavirus*, *Gemytondvirus*, and *Gemyvongvirus* [1].

The genus *Begomovirus* is composed of whitefly-transmitted species with one (=monopartite) or two (=bipartite) circular, ssDNA genomic component(s) with ≈2.6 kb that are encapsidated separately into twinned particles formed by two incomplete icosahedrons [2,4]. The begomovirus transmission is characterized as being non-propagative and circulative, and is carried out by members of the *Bemisia tabaci* (Hemiptera: Aleyrodidae) cryptic species complex [5]. The begomoviruses display a set of mechanisms for generating genetic variability such as mutation, recombination, and pseudo-recombination, which have a direct influence in the continuous emergence of new species that are often reported in this genus [6,7,8].

The tomato (*Solanum lycopersicum* L.) crop is grown year-round across major tropical and subtropical regions [9]. In Brazil, outbreaks of *Begomovirus* species in tomatoes became more intensively reported after the invasion of *B. tabaci* Middle East–Asia Minor 1 (MEAM 1 = biotype B) in the early 1990s [10]. The well-known biological attributes of *B. tabaci* MEAM 1 (viz. large host range, ability to transmit a wide range of viral species, and adaptation to distinct environmental conditions) facilitated the rapid dispersal of tomato-infecting begomoviruses across all major producing areas of the country [6]. Field surveys conducted afterward have revealed an extremely diverse complex of *Begomovirus* species (composed mainly by bipartite viruses), occurring in all Brazilian biomes. Currently, over 21 tomato-infecting *Begomovirus* species have been characterized in Brazil (Appendix A), and most of them are already accepted by the International Virus Taxonomy Committee (ICTV) [11]. In addition, begomoviruses initially reported in alternative weed hosts are also occasionally reported asinfecting tomatoes such as *Sida mottle virus* (SiMoV) and *Sida micrantha mosaic virus* (SiMMV) [12,13]. Currently, *Tomato severe rugose virus* (ToSRV; a bipartite species) and *Tomato mottle leaf curl virus* (ToMoLCV; a monopartite species) are the most widespread and economically important begomoviruses, with occurrence reported across all major tomato-producing regions, including central Brazil. The remaining viral species have an overall more restricted (sometimes endemic) geographic distribution [14].

The preferential strategy for begomovirus management in tomatoes is the employment of cultivars with genetic resistance/tolerance, since the use of insecticides for controlling viruliferous vector populations is neither efficient nor economically and environmentally sustainable [15,16]. Currently, eight resistance/tolerance genes/alleles to begomovirus have been characterized in *Solanum* (section *Lycopersicon*) germplasm: Ty-1 [17], Ty-2 [18], Ty-3 [19], Ty-4 [20], ty-5 [21], Ty-6 [22], tcm-1 [23], and tgr-1 [24]. The *Ty*-1 gene/locus introgressed from *Solanum chilense* LA 1969 (Zamir et al. 1994) is by far the most employed genetic factor in tomato breeding programs across the globe. In Brazil, cultivars carrying the *Ty*-1 gene have been widely used, mainly across producing regions in central Brazil [25,26]. The *Ty*-1 gene is located on chromosome 6 in a genomic region in repulsion phase linkage with resistance genes against other pathogens, including the *Mi*-1.2 gene that confers resistance in tomato to the three most important root-knot nematode species: *Meloidogyne incognita, Meloidogyne javanica,* and *Meloidogyne arenaria* [27,28]. Molecular markers for monitoring the presence of the *Ty*-1 gene/locus in tomato cultivars are now available [29,30,31].

The phenotypic expression of the *Ty*-1 gene is best described as a tolerance response [32], since plants harboring this factor allow for a mild manifestation of symptoms, mainly in the apical meristematic regions, which is followed by a progressive recovery as the plant growth/development advances [33]. This tolerant reaction is expressed against a relatively large number of monopartite and bipartite begomoviruses and it is related to the inhibition of viral movement, being more efficient under low inoculum conditions [17,33]. Genetic studies conducted by Verlaan et al., 2011 showed that the *Ty*-1 gene encodes an RNA-dependent RNA polymerase. Therefore, the *Ty*-1 gene is representing an entirely new class of disease resistance/tolerance genes that operates by intensifying the levels of transcriptional silencing of viral genes. More recent studies have shown that the *Ty*-1 gene can also confer resistance to *Beet curly top virus* (a viral species of the genus *Curtovirus*) in genetically transformed *Nicotiana benthamiana* plants [34]. However, no information is yet available about the effects of the *Ty*-1 gene on ssDNA viruses and subviral agents described in association with tomatoes in neotropical areas.

Next-generation sequencing (NGS) technologies have intensified the advances in elucidating many aspects of plant–microbe interactions by enabling the generation of a huge amount of low-cost sequence data of both hosts and pathogens [35]. Currently, metagenomic analyses with NGS are the best tools available for large-scale assessment of viral diversity under distinct environmental conditions [36,37,38]. NGS has contributed significantly to the sequencing of complete genomes as well as in detecting novel plant-associated viral species [36,37,38]. NGS technologies have also contributed to revealing the viral diversity associated with the tomato crop [39,40,41,42].

Due to the extreme variability of the neotropical tomato-infecting begomoviruses, it is possible that species and strains not yet identified can be emerging in this region. The increase in the crop acreage with tomato varieties and hybrids harboring the *Ty*-1 gene may represent a relevant selection factor on viral populations that could make them either more adapted or even capable of entirely overcoming this tolerance factor [25,26]. However, the viral diversity associated with the *Ty*-1 gene and other tomato resistance/tolerance factors have not yet been extensively studied. The complete sequence information of the DNA-A and DNA-B genomic segments generated by NGS provides large-scale assessment tools to study viral population diversity in the tomato–begomovirus pathosystem. In this context, the objective of the present work was to carry out metagenomic analyses aiming to reveal the diversity of *Begomovirus* species as well as other ssDNA viruses and subviral agents in tomato cultivars either lacking or harboring the *Ty*-1 gene in central Brazil.

## 2. Materials and Methods

### 2.1. Tomato Leaf Samples and Confirmation of the Presence/Absence of the Ty-1 Gene/Locus in the Genome of the Tomato Samples by Employing a Cleaved Amplified Polymorphic Sequence (CAPS) Marker System

Foliar samples (*n* = 107) of field-grown tomato cultivars/hybrids (with and without the *Ty*-1 tolerance gene) showing distinct degrees of begomovirus-like symptoms (viz. apical and interveinal chlorosis, yellow spots, golden mosaic, severe rugose mosaic, apical leaf deformation, and stunting) were collected from 2001 to 2016 across three geographic regions (Goiás State—GO, the Federal District—DF, and Minas Gerais State—MG). In our survey, the majority (over 95%) of the samples collected from plants displaying begomovirus-like symptoms (in both pools) were confirmed to be infected by one or more begomovirus species (see description in Section 2.2). 

In order to confirm the presence of the *Ty*-1 gene/locus, we performed PCR assays with the DNA of these 107 tomato leaf samples with positive begomovirus detection. We employed the primer pair UWTyF/UWTyR, which is capable of generating a CAPS marker linked to this tomato genomic region [29]. This codominant marker system is able to discriminate the dominant resistance allele (*Ty*-1) from the susceptible recessive allele (*ty*-1) after cleavage with the restriction enzyme *Taq* I [29]. In order to reveal these alternative alleles for the *Ty*-1 gene/locus, PCR products (amplicons) were cleaved with the enzyme *Taq* I for 2 h at a constant temperature of 65 °C. The products obtained after cleavage were analyzed in 1% agarose gels, stained in ethidium bromide, and visualized under ultraviolet light. 

### 2.2. Viral Isolates and Preliminary Confirmation of the Presence of Begomoviruses in the Tomato Leaf Samples

Each individual sample was subjected to total DNA extraction using a modified (high pH buffer) 2X CTAB + organic solvent protocol [43]. These samples/isolates were stored at −20 °C and they currently comprise a section of the begomovirus collection of the Plant Breeding Laboratory at CNPH (Brasília, DF, Brazil). The purified total DNA was subjected to polymerase chain reaction (PCR) assays aiming to confirm the presence of begomovirus(es) in these tomato leaf samples. Amplicons derived from a segment of the DNA-A component were obtained using the “universal” primer pairs PAL1v1978/ PAR1c496 [44] and BegomoAFor1′/‘BegomoARev1 [45], which produce two large and non-overlapping segments (≈1120 bp and ≈1205 bp, respectively). Amplicons derived from a segment of the DNA-B component (≈690 bp) were obtained using the “universal” primer pair PBL1v2040′/‘PCRc1 [44]. The obtained amplicons were analyzed in 1% agarose gels, stained in ethidium bromide, and visualized under ultraviolet light. Only samples displaying begomovirus-derived amplicons were selected for a subsequent enrichment of circular DNAs via rolling circle amplification and for next-generation sequencing (NGS; see sections below).

### 2.3. Enrichment via Rolling Circle Amplification of Circular DNA Molecules on Each Individual Sample

The virus*-*derived circular DNA molecules in the samples were selectively enriched by rolling circle amplification (RCA) assays [46]. After gel electrophoresis, the concentrations were adjusted via NanoVue Plus to 1 microgram per sample and then used to make up the two pools. The CAPS*-*characterized samples were then subdivided into two pools: one composed of DNAs of tomato plants without the Ty*-*1 (Table 1) gene (*n* = 64) and one composed of DNAs of tomato samples with the Ty-1 (Table 2) gene (*n* = 43).

### 2.4. Next-Generation Sequencing (NGS) of the Two Tomato DNA Pools and Analysis of the NGS-Derived Sequences

The sample pools (with and without Ty-1 gene) were subjected to high-performance sequencing in an Illumina platform with the HiSeq 2500 system (Macrogen Inc., Seoul, South Korea). The analyses were performed in paired ends reads of 100 nucleotides in length. The number of raw reads (filtered reads in parentheses) obtained for each pool were 16,227.547 (32,455.094) and 16,345.987 (32,691.974) for resistant and susceptible pools, respectively. The NGS-derived sequences were analyzed according to the following workflow: (1) elimination of low-quality reads, (2) assembly of the sequences using the program CLC Genomics Workbench 10, and (3) validation of the contigs via BLASTx and BLASTn algorithms by comparing with the ssDNA virus database of GenBank (https://www.ncbi.nlm.nih.gov/). The viral contigs were annotated and the trimmed reads were mapped back to the annotated genome using the tool “Map to reference” available in the Geneious 11.0 program [47]. The conserved regions/motifs present in the begomovirus genomes such as nonanucleotide, TATA box, stem loop, and iterons were also selectively analyzed [48]. Additionally, individual identification of the viruses was obtained in the NGS-derived dataset by using the SeqMan NGene Metagenomic sequence analysis software (DNAStar, Madison, WI, USA). Viral contigs were analyzed against the RefSeq viral database (NCBI) at a very high stringency conditions (minimum match percentage = 99%). The contig sequences were assembled by CLC Genomics Workbench and subsequently submitted to GenBank (see Section 3.1 below).

### 2.5. Design of a Collection of Viral Species-Specific PCR Primers for Detection in Individual Samples

For the confirmation of the viral species detected in each individual sample, specific PCR primers (for both DNA-A and DNA-B genomic segments) were designed in opposite and overlapping directions. Primer design was carried out on the basis of the consensus contigs obtained with the Geneious 11.0 program (Table 3). Virus specificity of the primers was double-checked in silico by using the Primer-Blast tool and in preliminary PCR assays using template DNA samples from a reference collection of the NGS-identified viral isolates.

### 2.6. Validation of NGS-Derived Information via PCR Assays with Virus-Specific Primers

PCR assays with the previously selected virus-specific primers (Table 3) were carried with in all 107 individual DNA samples. These assays were performed in order to validate the NGS results. PCR assays were carried with a total volume of 12.5 µL, containing 1.25 µL of *Taq* polymerase 10× buffer, 50 mM of MgCl_2_, 2.5 mM of dNTPs, 10 µM of each primer (forward and reverse), 100 ng of DNA, 8.0 µL of Milli-Q water, and 0.5 U *Taq* DNA polymerase. The reactions were amplified in a thermal cycler (Bio-Rad Laboratories, Hercules, CA, USA) programmed for 35 cycles with the following conditions: initial denaturation at 94 °C for 3 min, denaturation at 94 °C for 30 s, annealing (ranging from 46 to 60 °C, according to the primer pair employed; Table 3) for 45 s, extension at 72 °C for 3 min, and final extension at 72 °C for 10 min. The begomovirus-derived amplicons were observed to 1.5% agarose gel electrophoresis stained with ethidium bromide and visualized under UV light.

### 2.7. Sanger Dideoxy Sequencing Validation of Virus-Specific PCR Amplicons

Direct Sanger dideoxy sequencing reactions of positive virus-derived amplicons were carried out to double-check the viral diversity observed in a subset of individual samples. Sequencing reactions were performed at the Genomic Analysis Laboratory (at CNPH), employing the same virus-specific primer pairs (Table 3) in one ABI PRISM 3130 sequencer using the BigDye Terminator Cycle Sequencing Ready Reaction Kit version 3.1 protocol (Applied Biosystems, São Paulo–SP, Brazil). After contig assembling and quality evaluation, the obtained sequences were analyzed using the BLASTn algorithm. This tool was used to compare our sequences with the ones retrieved from the GenBank–NCBI public database (https://www.ncbi.nlm.nih.gov/), aiming to verify the sample-associated viral species. We adopted the current pairwise identities of 91% and 94% as the demarcation thresholds to identify *Begomovirus* species and strains, respectively [2].

## 3. Results

### 3.1. NGS Detection of Previously Reported Begomovirus Species in the Two Pools of Samples (with and without the Ty-1 Gene)

The assembled contig sequences of both pools that were characterized here and their corresponding GenBank accessions are presented in Table 4 and Table 5. The total number of reads per viral species/genomic component obtained in the pool without the *Ty*-1 gene is presented in Table 4, whereas the viral sequences obtained in pool from plants with the *Ty*-1 gene is presented in Table 5. After assembly, 19,487 contigs were obtained in the pool without the *Ty*-1 gene and 7045 contigs in the sample pool with the *Ty*-1 gene. ToSRV and *Tomato rugose mosaic virus* (ToRMV) displayed the two highest numbers of reads, indicating their relative predominance in the tomato samples with the *Ty*-1 gene. Ten begomoviruses were found in the pool without the *Ty*-1 gene. Interestingly, only the DNA-A component was recovered from well-characterized bipartite species, including *Bean golden mosaic virus* (BGMV), *Tomato common mosaic virus* (ToCmMV), and SiMMV. In the pool harboring the *Ty*-1 gene, the complete genomes of four previously described bipartite *Begomovirus* species were recovered *viz*. ToSRV, *Tomato chlorotic mottle virus* (ToCMoV), *Tomato golden vein virus* (TGVV), and ToRMV. The monopartite species ToMoLCV was also recovered in both pools. Some of the neotropical tomato-infecting *Begomovirus* species (included on the RefSeq database) displayed overall high identity levels (e.g., >97% identity in the case of the DNA-B component that is shared by the species ToSRV and ToRMV). The NGS sequences were then realigned against the viral genomes found using SeqMan NGene (99% stringency parameter) in order to minimize assembling artifacts. This allowed a more precise evaluation of the representation of each virus and subviral agents in the sequenced pools. The validation of the NGS results via PCR assays with virus-specific primers coupled with Sanger dideoxy sequencing was also a very important tool to verify the presence of each individual virus species described here.

### 3.2. NGS Detection of Putative Three Novel Begomovirus Species as Well as a New Alpha-Satellite Species and a Gemycircularvirus (Genomoviridae) in the Tomato Samples

We were also able to identify three putative new *Begomovirus* species (one in the pool with the *Ty*-1 gene and two species in the pool without the *Ty*-1 gene). In the pool of samples with the *Ty*-1 gene, a putative new virus (named here as species #1 = isolate DF-640) displayed a bipartite genome organization, having a DNA-A component with 2605 nts and a DNA-B component with 2625 nts. The putative new species #1 displayed the highest level of identity (85%) with *Tomato rugose yellow leaf curl virus* (TRYLCV) isolates. The isolate DF-640 was recovered from a field-grown tomato plant in the vicinities of Gama city (in the Federal District), with severe symptoms, indicating a putative increase in virulence in relation to the *Ty*-1 gene. However, we cannot exclude that these severe symptoms might have been induced by synergisms involving DF-640 and RNA viruses (since they were not investigated in the present study). Two putative new species were detected in the pool lacking the *Ty*-1 gene. The first one was tentatively named here as new species #2 (=isolate MG-378) and displayed only the DNA-A component with 2649 nts. *Tomato bright yellow mottle virus* (ToBYMoV) was the begomovirus with the highest identity level (84%) to the new species #2. Additional PCR assays were carried out using the isolate MG-378 as a template, but no amplicon for the putative cognate DNA-B component was recovered (data not shown), indicating that it is more likely a monopartite virus. The new species #3 is also more likely a monopartite begomovirus, having a DNA-A genome with 2636 nts. The new species #3 displayed the highest identity level (84%) to *Abutilon mosaic Brazil virus* (AbMV). In addition, a four genetically identical isolates of a novel alpha-satellite species and two isolates of a gemycircularvirus (Family: *Genomoviridae*) species were also detected exclusively in the pool of tomato samples without the *Ty*-1 gene (Table 4).

### 3.3. Confirmation via PCR Assays with Virus-Specific Primers and Sanger Dideoxy Sequencing of the Viral and Subviral ssDNA Species Present in each Individual Tomato Sample and Quantification of Mixed Infections

After carrying out PCR assays with virus-specific primers (Table 3) and Sanger sequencing, we were able to catalog all the viral and subviral ssDNA species present in each individual tomato sample comprising the two pools. In the samples of the pool without the *Ty*-1 gene, it was possible to confirm the presence of all *Begomovirus* species reported initially by the analyses of the NGS-derived results (Table 6). ToSRV was the most prevalent begomovirus, mainly in samples from Goiás (GO) State. In the samples of the pool with the *Ty*-1 gene, all species identified after NGS analyses were also confirmed via PCR assays with virus-specific primers (Table 7). In addition, it is important to highlight that the majority of the samples displayed mixed infections, with two to five viral species being simultaneously detected in a single tomato plant (Figure 1 and Figure 2).

## 4. Discussion

Over 286 viral species have been reported to infect tomatoes in Brazil and worldwide [5,49,50,51]. The emergence per se of a large number of novel species is somewhat expected since the begomoviruses display a well-known set of mechanisms for generating genetic variability such as mutation, recombination, and pseudo-recombination [6,7,52]. The scenario of immense begomovirus variability in the neotropics favors the emergence of new species, which can be intensified by the frequent occurrence of mixed infections. However, there is a surprisingly scarce amount of information quantifying the frequency of mixed infections of tomato plants by members of the neotropical *Begomovirus* species complex under natural conditions. Our NGS-derived results displayed a substantial number of the tomato samples with events of co-infection in both pools (with and without the *Ty*-1 gene). The simultaneous presence of distinct virus species detected in single plants ranged from two to up to five (Figure 1 and Figure 2). However, it is interesting to highlight that the *Ty*-1 gene did not have a significant impact on reducing the overall number of multiple viral infections, since samples with this genetic factor displayed non-significant differences when compared to samples without this gene (chi-square test = 6.5193; *p*-value = 0.1635, which was found to be not significant at *p* < 0.05).

In our study, in addition to the detection of *Begomovirus* species already reported in the neotropics, we were able to detect two putative new *Begomovirus* species in the samples without the *Ty*-1 and one novel *Begomovirus* species in a sample with the *Ty*-1 gene. The putative new species #1 (DF-640) displays all typical features of the New World bipartite begomoviruses, having a DNA-A with a size of 2605 nts and a DNA-B component with 2625 nts. The new species #1 was the only begomovirus found to be associated with severe (dwarf) symptoms in plants harboring the *Ty*-1 gene in our survey. The new species #1 displayed the highest identity level (85%) with *Tomato rugose yellow leaf curl virus* (TRYLCV). Only the DNA-A components were found in the putative new species #2 (=isolate MG-378) and in the new species #3 (=isolate GO-169), suggesting that both might be novel monopartite viruses. The new species #2 (2649 nts) displayed the highest identity (84%) with the *Tomato bright yellow mottle virus* (ToBYMoV), and the new species #3 (2636 nts) displayed the highest identity level (84%) with *Abutilon mosaic Brazil virus* (AbMBV). According to the current criteria for species demarcation in the genus *Begomovirus*, nucleotide identities of the DNA-A component that are less than 91% with the complete DNA-A genome of any other known begomovirus sequence will correspond to a new species [2]. The overall low number of samples detected with these putative new begomoviruses indicates that they may represent extremely rare emergence events of novel viral variants. Therefore, it is most likely that we were able to identify these emerging viruses here due to the enhanced analytical power provided by the NGS technology.

A single alpha-satellite isolate with 1321 nts was detected in four individual samples collected in distinct areas of the Federal District in plants lacking the *Ty*-1 gene. Alpha-satellite DNA molecules are subviral agents classified in the family *Alphasatellitidae* that have been found in association with *Begomovirus* [1,53]. The genera of alpha-satellites associated with the geminiviruses are found in the subfamily *Geminialphasatellitinae*, viz. *Ageyesisatellite*, *Clecrusatellite*, *Colecusatellite*, and *Gosmusatellite*. Nucleotide identity less than 88% (in comparison with complete sequences of the known alpha-satellites) is the criterion currently used for the classification of a new species within the family *Geminialphasatellitinae* [1,53]. The alpha-satellite isolates found in the present study showed the highest level of identity (81%) with other New World species that were found in association with bipartite begomoviruses in Brazil, Cuba, and Venezuela [54,55]. Thus, according to the demarcation within the subfamily, the alpha-satellite is more likely a new species, probably of the genus *Clecrusatellite*, which is composed of species found in association with bipartite *Begomovirus* from the New World [1,54,55]. This putative new alpha-satellite species was detected in constant association with two begomoviruses (TGVV and ToMLCV). Therefore, additional bioassays will be necessary to identify which associated *Begomovirus* species is able to transreplicate this novel alpha-satellite.

A plant-associated genomovirus 12, classified into the genus *Gemycircularvirus* (family *Genomoviridae*), was also detected in two tomato samples from the pool without the *Ty*-1 gene. Both isolates were collected in Leopoldo de Bulhões, Goiás (GO) State in 2004. These isolates displayed 98% identity to Capybara genomovirus 9 isolate cap1_561 (MK483081.1) from Brazil [56]. The gemycircularviruses have ssDNA, and some species of this genus have been reported in association with plants [57,58,59]. In Brazil, two gemycircularviruses were previously described in samples of *Momorcadia charantia* and *Euphorbia heterophylla* [60]. However, according to our knowledge, this is the first report of a gemycircularvirus associated with tomatoes in Brazil and worldwide.

In the present study, the complete DNA-A sequences of the begomoviruses BGMV, SiMMV, and ToCmMV were detected and subsequently confirmed in the individual samples via PCR assays and Sanger sequencing. BGMV was found in two samples collected in the Gama (DF) region in 2003 (isolates DF-045 and DF-046) and in one sample collected in Leopoldo de Bulhões (GO) (isolate GO-142). In fact, BGMV has been previously found in association with tomatoes in the Submédio São Francisco River valley in northeast Brazil [61]. However, this initial detection was carried out by using only DNA-A-specific probes without additional molecular characterization of the putative BGMV isolates [61]. Therefore, our work is the first to characterize tomato-infecting BGMV isolates. Interestingly, the DNA-B components of these BGMV isolates were not recovered from the samples, indicating that they might be using one alternative DNA-B component from another co-infecting species. Additional bioassays will be necessary to confirm this hypothesis.

SiMMV was detected in association with tomatoes in Goiás State (nine samples) and in the Federal District (three samples). SiMMV was already reported infecting tomatoes in Brazil [12]. All SiMMV isolates were found only in the pool without the *Ty*-1 gene (Table 5), suggesting virus-specific filtering effects by this genetic factor. It will be of interest to challenge plants harboring the *Ty*-1 gene with infectious SiMMV clones to confirm this potential high level of resistance to this pathogen. This work is now underway.

Somewhat surprising, only the DNA-A component of the bipartite species ToCmMV was detected in two samples of the pool without the *Ty*-1 gene (GO-023 and MG-388) collected in Luziânia (GO) and Viçosa (MG), respectively. The isolate GO-023 is a mixed infection with ToCMoV, and the isolate MG-388 is a mixed infection with ToSRV. The absence of the DNA-B component of ToCmMV also suggests that these isolates might be using this component of these co-infecting species. This hypothesis remains to be investigated. ToCmMV was initially reported infecting tomato plants in southeast Brazil [62,63]. However, ToCmMV was not yet reported in Goiás State (GO-023). Even though both ToCmMV isolates were found in the pool without the *Ty*-1 gene, there are reports indicating that this virus can replicate and cause mild symptoms in tomato plants carrying this tolerance factor (manuscript in preparation).

The DNA-A and DNA-B genome sequences of *Euphorbia yellow mosaic virus* (EuYMV) and *Cleome leaf crumple virus* (CILCrV) were also recovered in our NGS analyses only from samples without the *Ty*-1 gene. EuYMV and CILCrV were detected in samples from Minas Gerais State. EuYMV was first characterized as infecting the weed *E. heterophylla* [64], and CILCrV was first reported infecting the weed *Cleome affinis* [54]. However, according to our knowledge, this is the first report of these two viral species naturally associated with tomatoes. The detection of these two species reinforces the hypothesis that weeds can serve as a natural reservoir for begomoviruses that may be able to move and be able to infect cultivated plants such as tomatoes.

We found that the NGS analyses in combination with PCR assays with virus-specific primers and Sanger sequencing to be powerful tools that allow us to assess the relative prevalence of the predominant *Begomovirus* species in distinct geographic areas across central Brazil. ToSRV has been described as the predominant begomovirus species, as indicated by independent surveys carried out across all tomato-producing regions in Brazil [13,65,66]. In our study, ToSRV was also the predominant begomovirus, being found in 46 samples of the pool without the *Ty*-1 gene and in 26 samples in the pool harboring the *Ty*-1 gene. This ability of ToSRV isolates replicate in plants with the *Ty*-1 gene could also be considered as an additional factor that explains the overall predominance of this virus under Brazilian conditions.

ToSRV is the prevalent begomovirus in the central and meridional regions of Brazil [66,67], whereas ToMoLCV is predominant in the northeast region [66]. However, ToMoLCV is also often found in central Brazil [68], which was confirmed by our results. *Tomato chlorotic mottle virus* (ToCMoV) has already been reported across the northeast, southeast, and central Brazil [6,69]. However, our results indicated that besides the Federal District, a large number of ToCMoV-infected tomato samples were also identified in Goiás State. *Tomato golden vein virus* (TGVV) is commonly found in central Brazil [68], and our results are in agreement with this observation. *Tomato rugose mosaic virus* (ToRMV) is a recombinant viral species with genomic contributions of ToSRV and ToCMoV [69]. In accordance with our results, ToRMV was found to be one of the predominant viral species in the central region of Brazil, especially in the Goiás State [6,70].

The present work is the first exploratory study on the potential impact of the *Ty*-1 gene on the diversity of neotropical *Begomovirus* species. The majority of the samples obtained from tomato plants carrying the *Ty*-1 gene displayed mild symptoms restricted to the apical leaves. It was possible to observe putative filtering effects as well as gene-specific viral selection in samples with the *Ty*-1 gene, indicating a potential evolution of viral populations more adapted to this genetic factor. It would be interesting to know if the viruses detected in the apical mild symptoms in plants carrying the *Ty*-1 gene are indeed able to escape its effects or if the occurrence of multiple infections on these plants makes a more permissive cellular environment. However, an illustrative example is the isolate DF-640 that was the only one found in association with severe disease symptoms in a field-grown *Ty*-1 tomato plant. This strong susceptible-like reaction associated with the isolate DF-640 may indicate its potential ability to overcome the *Ty*-1 gene. Another possibility is that the isolate DF-640 may represent a singular “host switch” event that is not necessary associated with viral adaptation to the *Ty*-1 gene. The production of infectious DF-640 clones is now underway, and they will be used to verify this hypothesis. Nevertheless, it is well documented that the increase in the acreage of cultivars harboring resistance genes such as *Ty*-1 can result in strong selection forces towards more aggressive viral isolates, accelerating the change in the composition of the viral population and potentially culminating with the loss of effectivity of the source of resistance/tolerance [71,72,73,74,75]. Studies have also reported the “breakdown” of the resistance mediated by the *Ty*-2 gene caused by a strain of *Tomato yellow leaf curl virus* (TYLCV) [76] and by a strain of the *Tomato leaf curl Bangalore virus* (ToLCBV) in India [77].

Our preliminary set of analyses showed no unique (i.e., pool-specific) polymorphisms (data not shown). Several point mutations were found, but none of them were specific to the viruses present in the pool with or without the *Ty*-1 gene. Thus, another plausible explanation for some of the reported field events of *Ty*-1-mediated resistance/tolerance “breakdown” under Brazilian conditions could be related to some natural synergistic interactions with distinct group of viruses. In fact, it has been demonstrated that the *Ty*-1 gene does not confer resistance to major tomato-infecting RNA viruses such as *Tomato spotted wilt virus* (TSWV) and *Cucumber mosaic virus* (CMV) [78]. However, it has been demonstrated that RNA viruses can compromise resistance against begomoviruses, as previously shown during TYLCV and CMV co-infection, where there was a significant increase in TYLCV concentration that was due to the inhibition of the transcriptional gene silencing response by CMV 2b RNAi suppressor protein [78,79,80]. In the present work, it was not possible to assess the diversity of RNA viruses associated with the samples because the employed methodological approach did not allow us to analyze this group of viruses.

## 5. Conclusions

The results reported here provide useful information about the population dynamics of begomoviruses associated with tomato crops across three major tomato-producing regions of central Brazil in the last decade. Our work is in agreement with the notion that NGS is a powerful strategy to identify not only known viruses but also novel viruses and subviral agents. The present work is the first step towards the elucidation of the potential begomovirus adaptation to the tolerance factor *Ty*-1. However, in order to carry out a more precise study on the selective impact of the *Ty*-1 locus on begomovirus diversity and evolution, a distinct experimental strategy would probably be more appropriate, since our analysis was conducted on samples collected in different regions of a large country, in different years and from tomato plants grown in different microenvironmental situations. Therefore, it is possible that these variables (geographic area, climate, and year) can generate some biases that may not allow us to estimate the actual effect of the *Ty*-1 gene. For this purpose, the analysis could be more appropriately conducted on samples collected from experimental plots cultivated with tomato isolines with and without the *Ty*-1 gene. On the other side, our ecologically oriented approach allowed us to carry out a more ample exploration of an array of environments, which may enhance the opportunity to detect a larger number of yet undescribed viral species associated with the tomato crop. Even though with a slightly different number of evaluated samples in the pools with (*n* = 43) and without (*n* = 64) the *Ty*-1 gene, virus-specific PCR assays and Sanger sequencing validations of NGS**-**derived data indicated greater diversity (14 versus 6 species) in samples lacking this gene. Moreover, two novel *Begomovirus* species, one gemycircularvirus (*Genomoviridae*) and one alpha-satellite were identified exclusively in samples without *Ty*-1, whereas a novel begomovirus was found exclusively in the *Ty*-1 gene pool. These results indicated a potential viral adaptation to this tolerance factor, as well as virus-specific filtering effects of the *Ty*-1 on a subset of single-stranded DNA viruses and subviral agents. However, these hypotheses will be better tested with tomato isolines (with and without the *Ty*-1 gene) after controlled experiments employing infectious clones.

## Figures and Tables

**Figure 1 viruses-12-00819-f001:**
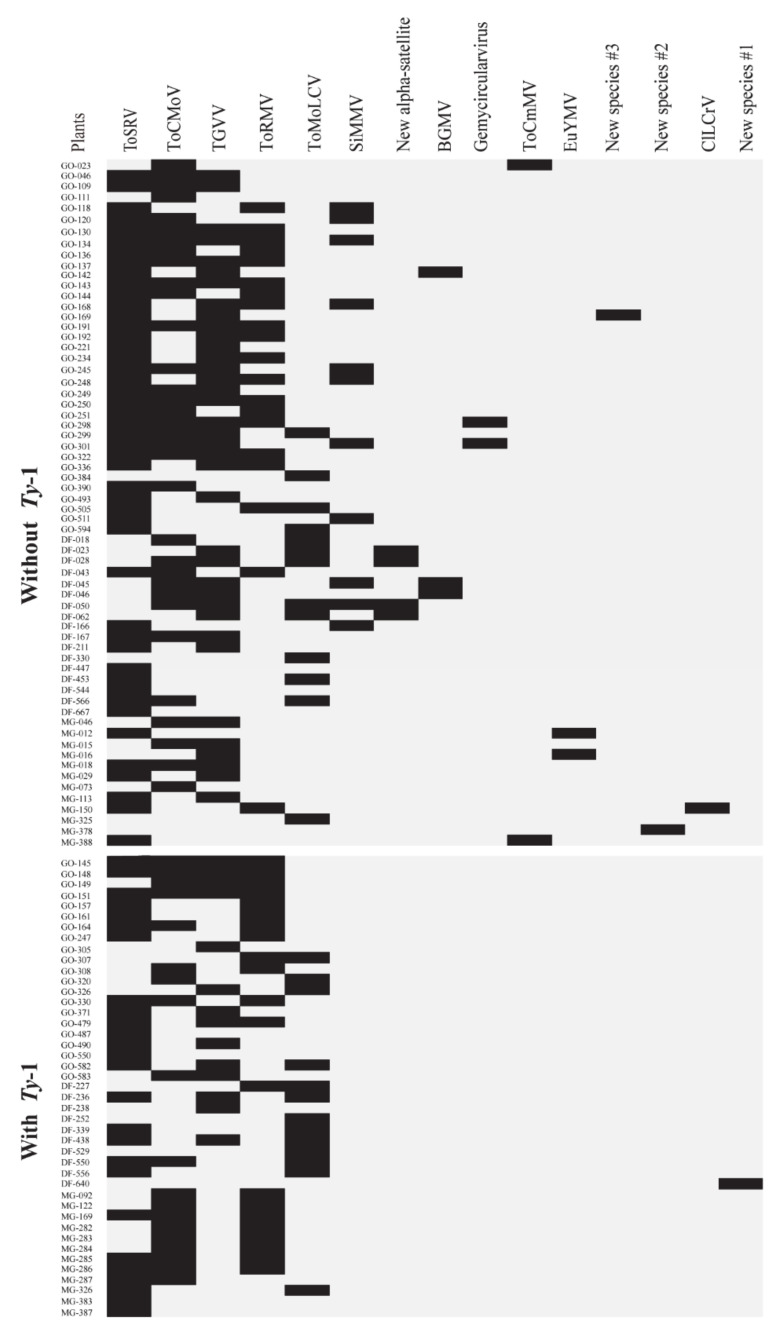
Frequency and relative predominance of *Begomovirus* species and single-stranded DNA viruses detected with Illumina Hiseq sequencing of tomato samples with (*n* = 43) and without (*n* = 64) the *Ty*-1 gene. Results were validated by PCR assays with virus-specific primers and by Sanger dideoxy sequencing. Viruses found were *Tomato severe rugose virus* (ToSRV); *Tomato golden vein virus* (TGVV); *Tomato chlorotic mottle virus* (ToCMoV); *Tomato rugose mosaic virus* (ToRMV); *Tomato mottle leaf curl virus* (ToMoLCV); *Sida micrantha mosaic virus* (SiMMV); *Bean golden mosaic virus* (BGMV); *Tomato common mosaic virus* (ToCmMV); *Euphorbia yellow mosaic virus* (EuYMV); and *Cleome leaf crumple virus* (CILCrV). A new alpha-satellite species and three putative novel begomovirus species (=new species #1, new species #2, and new species #3) were also detected. Black bars in each line indicate the presence of a given virus in a given individual sample = isolates (left column). Isolate with **GO** abbreviation = isolates collected in Goiás State; **DF** abbreviation = isolates collected in the Federal District; and **MG** abbreviation = isolates collected in Minas Gerais State, in central Brazil.

**Figure 2 viruses-12-00819-f002:**
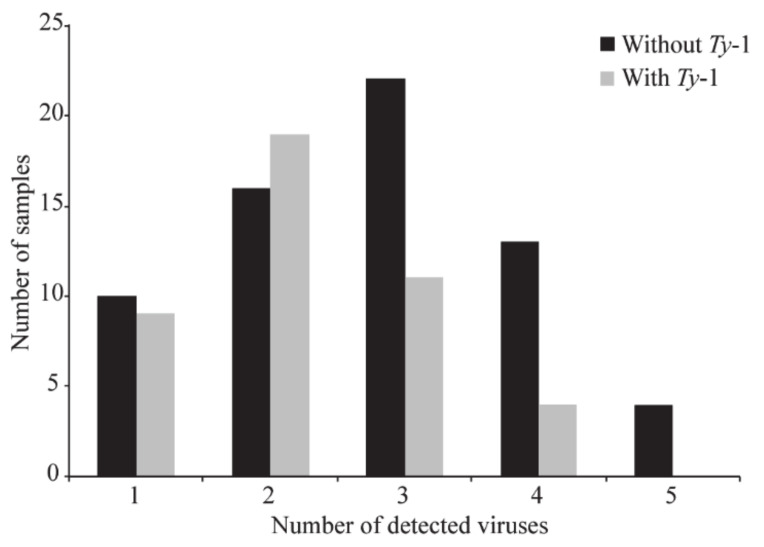
Number of samples displaying single and mixed (ranging from two to five viruses per sample) infections with *Begomovirus* species and single-stranded DNA viruses detected with Illumina Hiseq sequencing of tomato samples with (*n* = 43) and without (*n* = 64) the *Ty*-1 gene. Results were validated by PCR assays with virus-specific primers and by Sanger dideoxy sequencing.

**Table 1 viruses-12-00819-t001:** Identification of 64 samples (=isolates) exhibiting begomovirus-like symptoms that were obtained from tomato plants without the *Ty*-1 gene/locus in central Brazil. Information is provided about the region where the isolate was collected, year of collection, and the respective isolate code.

Geographic Region	Year of Collection	Isolate Code
**Goiás State—GO**	2003	GO-023, GO-046, GO-109, GO-111, GO-118, GO-120, GO-130, GO-134, GO-136, GO-137, GO-142, GO-143, GO-144, GO-168, GO-169, GO-191, GO-192, GO-221, GO-244, GO-245, GO-248, GO-249, GO-250, GO-251
	2004	GO-298, GO-299, GO-301, GO-322, GO-336
	2006	GO-384, GO-390
	2011	GO-493
	2012	GO-505, GO-511
	2015	GO-594
**Federal District—DF**	2003	DF-018, DF-023, DF-028, DF-043, DF-045, DF-046, DF-050, DF-062
	2005	DF-166, DF-167, DF-211
	2010	DF-330
	2011	DF-447, DF-453
	2013	DF-544
	2014	DF-566
	2016	DF-667
**Minas Gerais State—MG**	2001	MG-046
	2002	MG-012, MG-015, MG-016, MG-018, MG-029
	2010	MG-073, MG-113, MG-150
	2012	MG-325
	2015	MG-378, MG-388

**Table 2 viruses-12-00819-t002:** Identification of 43 samples (=isolates) exhibiting begomovirus-like symptoms that were obtained from tomato plants harboring the *Ty*-1 gene/locus in central Brazil. Information is provided about the region where the isolate was collected, year of collection, and the respective isolate code.

Geographic Region	Year of Collection	Isolate Code
**Goiás State—GO**	2003	GO-145, GO-148, GO-149, GO-151, GO-157, GO-161, GO-164
	2004	GO-247, GO-305, GO-307, GO-308, GO-320, GO-326, GO-330
	2007	GO-371
	2010	GO-479, GO-487, GO-490
	2013	GO-550, GO-582, GO-583
**Federal District—DF**	2007	DF-227, DF-236, DF-238
	2008	DF-252
	2010	DF-339
	2011	DF-438
	2013	DF-529, DF-550, DF-556
	2016	DF-640
**Minas Gerais State—MG**	2010	MG-092, MG-122, MG-169, MG-282, MG-283, MG-284, MG-285, MG-286, MG-287
	2012	MG-326
	2015	MG-383, MG-387

**Table 3 viruses-12-00819-t003:** PCR primer pairs designed on the basis of next-generation sequencing (NGS)-derived viral consensus sequences for validation of the *Begomovirus* species, as well as single-stranded DNA viruses and subviral agents identified in the tomato DNA sample pools (with the *Ty*-1 gene versus without the *Ty*-1 gene). For = forward and Rev = reverse direction.

Viral Species	Primer Name	Sequence 5′–3′	Annealing Temperature (T ^°^C)
*Bean golden mosaic virus* (BGMV) DNA-A	BGMV-For	GTGCGTGAATCCATGACCGT	55
	BGMV-Rev	ATTCACGCACAGGGGAACG	
*Cleome leaf crumple virus* (CILCrV) DNA-A	CILCrV-A-For	GACTCGACGTTCTGTGGT	51
	CILCrV-A-Rev	TCCTAGTCGGGGCTCACT	
*Cleome leaf crumple virus* (CILCrV) DNA-B	CILCrV-B-For	TAGGAAAGCAAAACGAGAATGGAA	58
	CILCrV-B-Rev	GCTTTCCTAAATCGCAATTGATC	
*Tomato severe rugose virus* (ToSRV) DNA-A	ToSRV-For5.1	AGCGTCGTTAGCTGTCTGGCA	58
	ToSRV-Rev5	TGCCGCAGAAGCCTTGAACGCACCT	
*Tomato severe rugose virus* (ToSRV) DNA-B	ToSRV-B-For	AAACCCACACGAAAGCAGAGTTT	55
	ToSRV-B-Rev	CACCACGTCTATACATATTGTCCAGG	
*Euphorbia yellow mosaic virus* (EuYMV) DNA-A	EuYMV-A-R-For	GGGGTTCCAAGTCCAATAAAGATGA	52
	EuYMV-A-R-Rev	CAGACACCTTATATTTGCCGGATTC	
*Euphorbia yellow mosaic virus* (EuYMV) DNA-B	EuYMV-B-R-For	GCCGAGGATAGAGGACACCAA	60
	EuYMV-B-R-Rev	CCAGGCCCAAACGCATTATATTTTATC	
*Tomato chlorotic mottle virus* (ToCMoV) DNA-A	ToCMoV-A-For	TTTGGGCCGCTCTTTTGGG	47
	ToCMoV-A-Rev	CAAACTGAATGGGCCTTAAA	
*Tomato chlorotic mottle virus* (ToCMoV) DNA-B	ToCMoV-B-For	GTATTTGTTCTGGGTGCAATCATAAAAC	55
	ToCMoV-B-Rev	TTGTACTAATGACACATTATTCAATCACGA	
*Tomato golden vein virus* (TGVV) DNA-A	TGVV-A-For1	AAAGGAAGATAATTCAAATATAGGGA	51
	TGVV-A-Rev1	ATCTTCCTTTACTCACGTTCCTGAT	
*Tomato golden vein virus* (TGVV) DNA-B	TGVV-B-S-For	CCCACTTTCCATAACCTACATGAGA	55
	TGVV-B-S-For	GGAGAGAAAATTGATAAGATCGGCATC	
*Tomato mottle leaf curl virus* (ToMLCV) DNA-A	ToMoLCV-For	TGTGGTCCAGTCAATAAATG	47
	ToMoLCV-Rev	TGACTGGACCACATAGTAAA	
*Tomato common mosaic virus* (ToCmMV) DNA-A	ToCmMV-For1	ATTGCTCTCAACTTCTGTGC	54
	ToCmMV-Rev2	GCAATCCCTGGTGTCCTCAC	
*Tomato rugose mosaic virus* (ToRMV) DNA-A	ToRMV-A-For	TGAAAGTAATTTTGACCCAATC	52
	ToRMV-A-Rev	CAATTCATATGAGTTTTAGAGCAGC	
*Sida micrantha mosaic virus* (SiMMV) DNA-A	SiMMV-For	GATCTCGCTCCCCCCTCT	58
	SiMMV-Rev	AGATCGCACGACAACCAG	
*Plant*-*associated genomovirus* 2	Gemy-For	GCTCTGAATCAAATCTCGCTTACTTG	54
	Gemy-Rev	CGATGTTGATTGGTTGGAAGCAAA	
New *Begomovirus* Species #1 DNA-A	DF-640-A-For	GTTGACTGACATTTGCCTT	47
	DF-640-A-Rev	TGTCAGTCAACAATCTATACACA	
New *Begomovirus* Species #1 DNA-B	DF-640-B-For	GTTGTTTCAAGGGCGTCGAC	55
	DF-640-B-Rev	CAACATCAGACATCCAGCAATAATAAACT	
New *Begomovirus* Species #2 DNA-A	1ToBYMV-A-For	ATCCATGTCCTCGGCAGTCT	55
	1ToBYMV-A-Rev	TCACGCACAGAGGAACGC	
New *Begomovirus* Species #3 DNA-A	Abuti-A-For	GGACTCCAGGGGGCAAAA	55
	Abuti-A-Rev	AGTCCCGTCCGTACCACTTG	
Alpha-satellite	Alfa-For	TGGTGTCCTGGCTTATAT	46
	Alfa-Rev	GGCGGAGTCCTTTTTTTT	

**Table 4 viruses-12-00819-t004:** Viral circular, single-stranded DNA species (and their corresponding GenBank accessions) detected after Illumina Hiseq sequencing in the pool of tomato DNA samples lacking the *Ty*-1 gene.

Viral Species	No. of Reads	Size (nts)	GenBank Accession Numbers
*Bean golden mosaic virus* (BGMV) DNA-A	63,525	2626	MT214083
*Cleome leaf crumple virus* (CILCrV) DNA-A	566	2560	MN337873
*Cleome leaf crumple virus* (CILCrV) DNA-B	702	2664	MN337872
*Tomato severe rugose virus* (ToSRV) DNA-A	3,225,120	2593	MT214084
*Tomato severe rugose virus* (ToSRV) DNA-B	4,018,351	2572	MT214085
*Euphorbia yellow mosaic virus* (EuYMV) DNA-A	1122	2609	MN746971
*Euphorbia yellow mosaic virus* (EuYMV) DNA-B	1822	2579	MN746970
*Tomato chlorotic mottle virus* (ToCMoV) DNA-A	5,971,019	2620	MT214086
*Tomato chlorotic mottle virus* (ToCMoV) DNA-B	1,111,227	2600	MT214087
*Tomato golden vein virus* (TGVV) DNA-A	2,639,961	2562	MN928610
*Tomato golden vein virus* (TGVV) DNA-B	977,027	2512	MN928611
*Tomato mottle leaf curl virus* (ToMLCV) DNA-A	1,784,881	2632	MT214088
*Tomato common mosaic virus* (ToCmMV) DNA-A	1,070,674	2560	MT214089
*Tomato rugose mosaic virus* (ToRMV) DNA-A	3,267,808	2619	MT214090
*Tomato rugose mosaic virus* (ToRMV) DNA-B	4,742,730	2571	MT214091
*Sida micrantha mosaic virus* (SiMMV) DNA-A	1,221,062	2688	MT214092
*Plant-associated genomovirus* 2	119	2189	MT214094
*New Begomovirus Species #2* DNA-A	427,646	2649	MT214095
*New Begomovirus Species #3* DNA-A	2839	2636	MT214096
*New alpha-satellite*	155,793	1321	MT214093

**Table 5 viruses-12-00819-t005:** Viral circular, single-stranded DNA species (and their corresponding GenBank accessions) detected after Illumina Hiseq sequencing in the pool of tomato DNA samples harboring the *Ty*-1 gene.

Viral Species	N° of Reads	Size (nts)	GenBank Accession Numbers
*Tomato severe rugose virus* (ToSRV) DNA-A	7,181,771	2592	MT215001
*Tomato severe rugose virus* (ToSRV) DNA-B	5,782,296	2570	MT215002
*Tomato golden vein virus* (TGVV) DNA-A	2,358,838	2561	MN928612
*Tomato golden vein virus* (TGVV) DNA-B	1,401,684	2590	MN928613
*Tomato chlorotic mottle virus* (ToCMoV) DNA-A	4,519,040	2623	MT215003
*Tomato chlorotic mottle virus* (ToCMoV) DNA-B	811,733	2565	MT215004
*Tomato mottle leaf curl virus* (ToMLCV) DNA-A	2,644,606	2631	MT215005
*Tomato rugose mosaic virus* (ToRMV) DNA-A	7,964,942	2618	MT215006
*Tomato rugose mosaic virus* (ToRMV) DNA-B	5,780,864	2649	MT215007
*New Begomovirus Species* #1 DNA-A	1,270,494	2605	MN147863
*New Begomovirus Species* #1 DNA-B	84,022	2625	MN147864

**Table 6 viruses-12-00819-t006:** Relative frequency of *Begomovirus* and other circular single-stranded DNA viruses detected after Illumina Hiseq sequencing of 63 tomato DNA samples lacking the *Ty*-1 gene.

Viral Species* Followed by the Respective Number of Occurrences in Each Region	Goiás State (GO)	Federal District (DF)	Minas Gerais State (MG)
**ToSRV** **(32 + 9 + 5) = 46**	GO-046, GO-109, GO-118, GO-120, GO-130, GO-134, GO-136, GO-137, GO-142, GO-143, GO-144, GO-168, GO-169, GO-191, GO-192, GO-221, GO-244, GO-245, GO-248, GO-249, GO-250, GO-251, GO-298, GO-299, GO-301, GO-322, GO-336, GO-390, GO-493, GO-505, GO-511, GO-594	DF-043, DF-166, DF-167, DF-211, DF-447, DF-453, DF-544, DF-566, DF-667	MG-012, MG-018, MG-029, MG-150, MG-388
**TGVV** **(23 + 8 + 5) = 36**	GO-046, GO-109, GO-130, GO-134, GO-137, GO-142, GO-143, GO-168, GO-169, GO-191, GO-192, GO-221, GO-244, GO-245, GO-248, GO-249, GO-250, GO-298, GO-299, GO-301, GO-322, GO-336, GO-493	DF-023, DF-028, DF-045, DF-046, DF-050, DF-062, DF-167, DF-211	MG-015, MG-016, MG-018, MG-029, MG-046
**ToCMoV** **(21 + 8+5) = 34**	GO-023, GO-046, GO-109, GO-111, GO-120, GO-130, GO-134, GO-136, GO-137, GO-143, GO-144, GO-191, GO-245, GO-249, GO-250, GO-251, GO-298, GO-299, GO-301, GO-322, GO-390	DF-018, DF-028, DF-043, DF-045, DF-046, DF-050, DF-167, DF-566	MG-015, MG-018, MG-046, MG-073, MG-150
**ToCmMV** **(1 + 0 + 1) = 2**	GO-023	---	MG-388
**BGMV** **(1 + 2 + 0) = 3**	GO-142	DF-045, DF-046	---
**CILCrV** **(0 + 0 + 1) = 1**	---	---	MG-150
**EuYMV** **(0 + 0 + 2) = 2**	---	---	MG-012, MG-016
**ToMLCV** **(4 + 8 + 1) = 13**	GO-299, GO-384, GO-505, GO-594	DF-018, DF-023, DF-028, DF-050, DF-062, DF-330, DF-453, DF-566	MG-325.
**ToRMV** **(19 + 1 + 1) = 21**	GO-109, GO-118, GO-130, GO-134, GO-136, GO-137, GO-143, GO-144, GO-168, GO-191, GO-192, GO-244, GO-248, GO-250, GO-251, GO-298, GO-322, GO-336, GO-505	DF-043	MG-150
**SiMMV** **(8 + 3 + 0) = 11**	GO-118, GO-120, GO-134, GO-168, GO-245, GO-248, GO-301, GO-511	DF-045, DF-050, DF-166	---
***Plant-associated genomovirus 2*** **(2 + 0 + 0) = 2**	GO-298, GO-301	---	---
**Alpha-satellite** **(0 + 4 + 0) = 4**	---	DF-023, DF-028, DF-050, DF-062	---
**New *Begomovirus* species #2** **(0 + 0 + 1) = 1**	---	---	MG-378
**New *Begomovirus* species #3** **(1 + 0 + 0) = 1**	GO-169	---	---

* ToSRV = Tomato severe rugose virus, TGVV = Tomato golden vein virus, ToCMoV = Tomato chlorotic mottle virus, ToCmMV = Tomato common mosaic virus, BGMV = Bean golden mosaic virus, CILCrV = Cleome leaf crumple virus, EuYMV = Euphorbia yellow mosaic virus, ToMLCV = Tomato mottle leaf curl virus, ToRMV = Tomato rugose mosaic virus, and SiMMV = Sida micrantha mosaic virus.

**Table 7 viruses-12-00819-t007:** Relative frequency of *Begomovirus* and other circular single-stranded DNA viruses in association with 43 tomato DNA samples harboring the *Ty*-1 gene detected after Illumina Hiseq sequencing.

Viral Species* Followed by the Respective Number of Occurrences in Each Region	Goiás State (GO)	Federal District (DF)	Minas Gerais State (MG)
**ToSRV** **(14 + 5 + 7) = 26**	GO-145, GO-148, GO-151, GO-157, GO-161, GO-164, GO-247, GO-330, GO-371, GO-479, GO-487, GO-490, GO-550, GO-582	DF-236, DF-339, DF-438, DF-550, DF-556	MG-169, MG-285, MG-286, MG-287, MG-326, MG-383, MG-387
**TGVV** **(12 + 3 + 0) = 15**	GO-145, GO-148, GO-149, GO-151, GO-305, GO-320, GO-326, GO-371, GO-479, GO-490, GO-582, GO-583	DF-236, DF-238, DF-438	---
**ToCMoV** **(12 + 1 + 9) = 22**	GO-145, GO-148, GO-149, GO-305, GO-320, GO-326, GO-330, GO-371, GO-479, GO-490, GO-582, GO-583	DF-550	MG-092, MG-122, MG-169, MG-282, MG-283, MG-284, MG-285, MG-286, MG-287
**ToMLCV** **(4 + 8 + 1) = 13**	GO-307, GO-320, GO-326, GO-582	DF-227, DF-236, DF-252, DF-339, DF-438, DF-529, DF-550, DF-556	MG-326
**ToRMV** **(13 + 1 + 8) = 22**	GO-145, GO-148, GO-149, GO-151, GO-157, GO-161, GO-164, GO-247, GO-307, GO-308, GO-320, GO-330, GO-479	DF-227	MG-092, MG-122, MG-169, MG-282, MG-283, MG-284, MG-285, MG-286
**New *Begomovirus* species #1** **(0 + 1 + 0) = 1**	---	DF-640	---

* ToSRV = Tomato severe rugose virus, TGVV = Tomato golden vein virus, ToCMoV = Tomato chlorotic mottle virus, and ToRMV = Tomato rugose mosaic virus.

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
