# Peer review of "Metagenomics of Neotropical Single-Stranded DNA Viruses in Tomato Cultivars with and without the Ty-1 Gene"

_viruses, 2020, doi:10.3390/v12080819_

Round 1

Reviewer 1 Report

In this work, the authors have studied the diversity of ssDNA viruses in genotypes of tomato harboring or not the tolerance gene Ty-1 using Illumina HTS. They found more virus species infecting tomato samples lacking the Ty-1 gene, including two novel begomoviruses (new species #2 and #3) and a new alpha-satellite. Interestingly, they have found another undescribed begomovirus that infects only the Ty-1 samples (new species #1). This work is an example of the use of HTS to identify not only known viruses but also novel viruses and subviral agents and a first step to study their adaptation to their hosts. One of the strengths of this work is the long-period survey of tomato samples, from 2001 to 2016 in three different Neotropical regions and the high number of samples analyzed (107).

I recommend this work for publication after some review:

  1. The two pools for NGS were from 64 (Ty-1 -) and 43 (Ty-1 +) tomato samples respectively (107 in total). However, prior to the pooling, the authors selected the samples infected by any begomovirus using universal primers and PCR. Does this mean that all 107 samples were infected or that those 107 samples were the infected ones among a higher number of analyzed samples? In the latter case, I believe they should indicate the number of samples analyzed to give a percentage of the incidence of begomoviruses in that survey.
  2. I believe that the authors should include some information regarding the Illumina sequencing: SE- or PE-reads; length of the sequencing reads; number of raw reads obtained for each pool and number of reads after filtering by quality and used for the assembly and the later remapping.
  3. It is not clear if the sequences submitted to GenBank are the contigs assembled by CLC Genomics Workbench, the consensus sequences obtained with Geneious or if they are the contigs assembled from the HTS data and corrected by Sanger sequencing.
  4. I do not understand the sentence on line 253 “This implies that some of our reads (Tables 4 and 5) were most likely counted more than once”. They have realigned the reads against the viral genomes found, so they really know the degree of redundancy in the sequenced pools.
  5. Comparing the sequences of ToMSDV (new virus species #1, MT215017 and MT215018) in the supplementary file with the sequences of ToMSDV in GenBank (MN147864.1 and MN147863.1), the percentages of identity are 99.92% and 92.21% for DNA-B and DNA-A, respectively. The sequences in the database were submitted by the same authors in 2019, I doubt, however, whether this paper is the first report of this virus species since the sequences are different. Indeed, it is the same isolate DF-640. Please, clarify this point.
  6. It is not clear to me if, apart from the new alpha-satellite, the authors have been able to identify other alpha-satellites in the samples. In discussion line 384 they said that they have found four alpha-satellite isolates but according to their results it seems that it is the same isolate in 4 different samples.
  7. The authors mention several times in the text that recombination is a common mechanism to generate genetic variability in begomoviruses, so I am wondering if the authors have searched for recombinant virus variants and how.
  8. This study is very complete, but in my opinion, this work can be improve if the authors perform a phylogenetic analysis to decipher the relationship of the viral isolates found in this work between tomatoes lacking and harboring the Ty-1 gene and with other isolates in public databases. I believe it would be interesting to perform that study at least for the common virus species found in the two pools: ToSRV, TGVV, ToCMoV, ToMLCV and ToRMV.
  9. Regarding the new virus species #1, I found very interesting that it is the only virus infecting the sample in which it was detected. Also, if I understood well, in the abstract, the authors said that this virus was the only one inducing severe symptoms in Ty-1 samples. I believe this is something relevant to further comment in the discussion section.
  10. Regarding the discussion, I believe it is very long and sometimes it is more an introduction that a discussion. I recommend its overall revision.

Author Response

Suggestions

Decisions/explanations

1.            The two pools for NGS were from 64 (Ty-1 -) and 43 (Ty-1 +) tomato samples respectively (107 in total). However, prior to the pooling, the authors selected the samples infected by any begomovirus using universal primers and PCR. Does this mean that all 107 samples were infected or that those 107 samples were the infected ones among a higher number of analyzed samples? In the latter case, I believe they should indicate the number of samples analyzed to give a percentage of the incidence of begomoviruses in that survey.

The suggested information was incorporated into this new version of the manuscript. In this work we selected 107 samples previously confirmed positive by PCR using primers PAL-PAR (Rojas et al., 1993).

Page 3, lines 110-116. In the beginning of the Material and Methods section we added:

Foliar samples (n=107) of field–grown tomato cultivars/hybrids (with and without the Ty–1 tolerance gene) showing distinct degrees of begomovirus–like symptoms (viz. apical and interveinal chlorosis, yellow spots, golden mosaic, severe rugose mosaic, apical leaf deformation, and stunting) were collected from 2001 to 2016 across three geographic regions (Goiás State–GO, the Federal District–DF, and Minas Gerais State–MG). In our survey, the majority (over 95%) of the samples collected from plants displaying begomovirus–like symptoms (in both pools) were confirmed to be infected by one or more begomovirus species (see description in 2.2).

2.            I believe that the authors should include some information regarding the Illumina sequencing: SE- or PE-reads; length of the sequencing reads; number of raw reads obtained for each pool and number of reads after filtering by quality and used for the assembly and the later remapping.

Suggestion accepted.

Page 6, lines 167-170:

We added: … “The analyses were performed in Paired-Ends (PE) in reads of 100 nucleotides of length. The number of raw reads (filtered reads in parentheses) obtained for each pool were: 16,227.547 (32,455.094) and 16,345.987 (32,691.974) for resistant pool and susceptible pools respectively”.

3.            It is not clear if the sequences submitted to GenBank are the contigs assembled by CLC Genomics Workbench, the consensus sequences obtained with Geneious or if they are the contigs assembled from the HTS data and corrected by Sanger sequencing.

Suggestion accepted.

Page 6, lines 181-182: We added:

“Sequences of contigs assembled by CLC Genomics Workbench were submitted to GenBank.”

We also added in Page 6, lines 218 and 219, … “These partial sequences obtained with specific primers were submitted to GenBank.”

4.            I do not understand the sentence on line 253 “This implies that some of our reads (Tables 4 and 5) were most likely counted more than once”. They have realigned the reads against the viral genomes found, so they really know the degree of redundancy in the sequenced pools.

We have deleted the sentence “This implies that some of our reads (Tables 4 and 5) were most likely counted more than once” since it is bringing confusion to text. 

The NGS sequences were realigned against the viral genomes found using SeqMan NGene (99% stringency parameter) in order to minimize assembling artifacts. This allowed to a more precise evaluation of the representation of each virus and subviral agents in the sequenced pools.

5.            Comparing the sequences of ToMSDV (new virus species #1, MT215017 and MT215018) in the supplementary file with the sequences of ToMSDV in GenBank (MN147864.1 and MN147863.1), the percentages of identity are 99.92% and 92.21% for DNA-B and DNA-A, respectively. The sequences in the database were submitted by the same authors in 2019, I doubt, however, whether this paper is the first report of this virus species since the sequences are different. Indeed, it is the same isolate DF-640. Please, clarify this point.

Sorry about this confusion. The correct sequences for DNA-A of DF-640 is MN147863 and for DNA-B is MN147864.

Page 10, Table 5

6.            It is not clear to me if, apart from the new alpha-satellite, the authors have been able to identify other alpha-satellites in the samples. In discussion line 384 they said that they have found four alpha-satellite isolates but according to their results it seems that it is the same isolate in 4 different samples.

This point was clarified into the new version of the manuscript. We added this information in both Result and Discussion sections.

Results section

Page 10, lines 274-276.

In addition, a four genetically identical isolates of a novel alpha–satellite species and two isolates of a gemycircularvirus (Family: Genomoviridae) species were also detected exclusively in the pool of tomato samples without the Ty–1 gene (Table 4).

Discussion section

Page 16, lines 368-383.

A single alpha–satellite isolate (with 1,321 nts) was detected in four individual samples collected in distinct areas of the Federal District in plants lacking the Ty–1 gene. Alpha–satellite DNA molecules are subviral agents classified in the family Alphasatellitidae that have been found in association with Begomovirus [1, 53]. The genera of alpha–satellites associated with the geminiviruses are found in the subfamily Geminialphasatellitinae, genus Ageyesisatellite, Clecrusatellite, Colecusatellite and Gosmusatellite. Nucleotide identity less than 88% (in comparison with complete sequences of the known alpha–satellites) is the criterion currently used for the classification of a new species within the family Geminialphasatellitinae [1, 53]. The alpha–satellite isolates found in the present study showed the highest level of identity (81%) with other New World species that were found in association with bipartite begomoviruses in Brazil, Cuba, and Venezuela [54, 55]. Thus, according to the demarcation within the subfamily, the alpha–satellite is more likely a new species, probably of the genus Clecrusatellite, which is composed by species found in association with bipartite Begomovirus from the New World [1, 54, 55]. This putative new alpha–satellite species was detected in constant association with two begomoviruses (TGVV and ToMLCV). Therefore, additional bioassays will be necessary to identify which associated Begomovirus species is able to transreplicate this novel alpha–satellite.

7.            The authors mention several times in the text that recombination is a common mechanism to generate genetic variability in begomoviruses, so I am wondering if the authors have searched for recombinant virus variants and how.

Recombination events occurring in begomovirus species have been studied by researchers from Brazil. Some papers have been published. For instance, Lima et al. (2013) and Silva et al. (2014).

Lima ATM, Sobrinho RR, Gonzalez-Aguilera J, Rocha CS, Silva SJC, Xavier CAD, Silva FN, Duffy S, Zerbini FM. Synonymous site variation due to recombination explains higher genetic variability in begomovirus populations infecting non-cultivated hosts. J Gen Virol. 2013; 94:418–431.

Silva, F. N., Lima, A. T., Rocha, C. S., Castillo-Urquiza, G. P., Alves-Júnior, M., & Zerbini, F. M. (2014). Recombination and pseudorecombination driving the evolution of the begomoviruses Tomato severe rugose virus (ToSRV) and Tomato rugose mosaic virus (ToRMV): two recombinant DNA-A components sharing the same DNA-B. Virology journal, 11(1), 66.

We have performed analyses for recombination in sequences obtained for three new species using RDP4 software (Martin et al., 2015). However, details about this result are included in other papers about molecular and biological characterization. We detected recombination events for MG-378 and DF-640. For GO-169 no event of recombination was detected.

Martin DP, Murrell B, Golden M, Khoosal A, Muhire B (2015) RDP4: Detection and analysis of recombination patterns in virus genomes. Virus Evol 1: vev003.

8.            This study is very complete, but in my opinion, this work can be improved if the authors perform a phylogenetic analysis to decipher the relationship of the viral isolates found in this work between tomatoes lacking and harboring the Ty-1 gene and with other isolates in public databases. I believe it would be interesting to perform that study at least for the common virus species found in the two pools: ToSRV, TGVV, ToCMoV, ToMLCV and ToRMV.

This is a very good suggestion. However, it is a broad topic that we believe that may deserve an additional study.

9.            Regarding the new virus species #1, I found very interesting that it is the only virus infecting the sample in which it was detected. Also, if I understood well, in the abstract, the authors said that this virus was the only one inducing severe symptom in Ty-1 samples. I believe this is something relevant to further comment in the discussion section.

This information was added to the Discussion section in the following context (lines 444-453).

Page 18, lines 445-446 and 452-453

The present work is the first exploratory study on the potential impact of the Ty–1 gene on the diversity of Neotropical Begomovirus species. The majority of the samples obtained from tomato plants carrying the Ty–1 gene displayed mild symptoms restricted to the apical leaves. It was possible to observe putative filtering effects as well as gene–specific viral selection in samples with the Ty–1 gene, indicating a potential evolution of viral populations more adapted to this genetic factor. It would be interesting to know if the viruses detected in the apical mild symptoms in plants carrying the Ty–1 gene are indeed able to escape its effects or if the occurrence of multiple infections on these plants makes a more permissive cellular environment. However, an illustrative example is the isolate DF–640 that was the only one recovered from a field–grown tomato plant carrying the Ty–1 gene displaying severe disease symptoms.

10.          Regarding the discussion, I believe it is very long and sometimes it is more an introduction that a discussion. I recommend its overall revision.

Discussion was substantially shortened in this new version of the manuscript. Sections/paragraphs that were shortened in the text are highlighted in red.

Reviewer 2 Report

Formal remarks:

  1. Although long-term sampling and massive sequence data analyses were performed, the text is relatively extensive regarding the volume of presented results. It may be slightly shortened to improve the readers comfort, e.g. by deleting some irrelevant passages (lines 104-115 - not important in the context of the work, lines 234-249 - only repeat the data from the tables, several parts from the Discussion only mechanically repeat facts mentioned in the Results). 
  2. line 36 - Genomoviridae - correctly in italic
  3. line 160 - Inoue-Nagata et al., 2004 - wrong format of the citation, moreover, this reference is not present in the list
  4. lines 198-201 - instead of complicated description of the PCR mix (volumes and stock concentrations of components) I recommend to mention simply the reaction volume and final concentrations
  5. In the text there is no reference to the supplementary data

Objective problems:

  1. The mutual similarity (sequence identity) of the four alphasatellite isolates should be mentioned
  2. The BLAST results of particular sequences are not actual and have to be updated (as verified using the supplementary sequence data) 

Author Response

Suggestion

Decision/explanation

1.            Although long-term sampling and massive sequence data analyses were performed, the text is relatively extensive regarding the volume of presented results. It may be slightly shortened to improve the readers comfort, e.g. by deleting some irrelevant passages (lines 104-115 - not important in the context of the work, lines 234-249 - only repeat the data from the tables, several parts from the Discussion only mechanically repeat facts mentioned in the Results).

The Introduction and the overall Discussion were substantially shortened in this new version of the manuscript. Sections that were shortened in the text are highlighted in red.

2.            line 36 - Genomoviridae - correctly in italic

Suggestion accepted.

Page 1, line 35.

We replaced Genomoviridae by Genomoviridae.

3.            line 160 - Inoue-Nagata et al., 2004 - wrong format of the citation, moreover, this reference is not present in the list

Suggestion accepted.

Page 4, line 146.

 We replaced Inoue-Nagata et al., 2004 by [46].

We also added the reference [46].

4.            lines 198-201 - instead of complicated description of the PCR mix (volumes and stock concentrations of components) I recommend to mention simply the reaction volume and final concentrations

Suggestion accepted.

 Page 6, lines 197-199.

 We replaced by:

PCR assays were carried with a total volume of 12.5 μL, containing 1.25 μL of Taq polymerase 10X buffer, 50 mM of MgCl2, 2.5 mM of dNTPs, 10 μM of each primer (forward and reverse), 100 ng of DNA, 8.0 μL of Milli-Q® water and 0,5 U Taq DNA polymerase.

5.            In the text there is no reference to the supplementary data

We have sent the sequences in a separate file, since they will not yet available on the GenBank. If is necessary, we can include this table as supplementary material.

Objective problems:

1.            The mutual similarity (sequence identity) of the four alphasatellite isolates should be mentioned

This point was mentioned and clarified into the new version of the manuscript. Thank you by this comment. We added this information in both Result and Discussion sections.

Results section

Page 10, lines 274-276.

In addition, a four genetically identical isolates of a novel alpha–satellite species and two isolates of a gemycircularvirus (Family: Genomoviridae) species were also detected exclusively in the pool of tomato samples without the Ty–1 gene (Table 4).

Discussion section

Page 16, lines 368-383.

A single alpha–satellite isolate (with 1,321 nts) was detected in four individual samples collected in distinct areas of the Federal District in plants lacking the Ty–1 gene. Alpha–satellite DNA molecules are subviral agents classified in the family Alphasatellitidae that have been found in association with Begomovirus [1, 53]. The genera of alpha–satellites associated with the geminiviruses are found in the subfamily Geminialphasatellitinae, genus Ageyesisatellite, Clecrusatellite, Colecusatellite and Gosmusatellite. Nucleotide identity less than 88% (in comparison with complete sequences of the known alpha–satellites) is the criterion currently used for the classification of a new species within the family Geminialphasatellitinae [1, 53]. The alpha–satellite isolates found in the present study showed the highest level of identity (81%) with other New World species that were found in association with bipartite begomoviruses in Brazil, Cuba, and Venezuela [54, 55]. Thus, according to the demarcation within the subfamily, the alpha–satellite is more likely a new species, probably of the genus Clecrusatellite, which is composed by species found in association with bipartite Begomovirus from the New World [1, 54, 55]. This putative new alpha–satellite species was detected in constant association with two begomoviruses (TGVV and ToMLCV). Therefore, additional bioassays will be necessary to identify which associated Begomovirus species is able to transreplicate this novel alpha–satellite.

2.            The BLAST results of particular sequences are not actual and have to be updated (as verified using the supplementary sequence data)

We could not understand quite well this comment since we have run the BLAST analyses and the results are identical.